# Prediction of pregnancy-related hypertensive disorders using metabolomics: a systematic review

Jussara Mayrink, Debora F Leite, Guilherme M Nobrega ⓘ , Maria Laura Costa, Jose Guilherme Cecatti ⓘ

Department of Obstetrics and Gynecology, State University of Campinas Faculty of Medical Sciences, Campinas, Brazil

**Correspondence to**
Professor Jose Guilherme Cecatti; cecatti@unicamp.br

## ABSTRACT

**Objective** To determine the accuracy of metabolomics in predicting hypertensive disorders in pregnancy.

**Design** Systematic review of observational studies.

**Data sources and study eligibility criteria** An electronic literature search was performed in June 2019 and February 2022. Two researchers independently selected studies published between 1998 and 2022 on metabolomic techniques applied to predict the condition; subsequently, they extracted data and performed quality assessment. Discrepancies were dealt with a third reviewer. The primary outcome was pre-eclampsia. Cohort or case–control studies were eligible when maternal samples were taken before diagnosis of the hypertensive disorder.

**Study appraisal and synthesis methods** Data on study design, maternal characteristics, how hypertension was diagnosed, metabolomics details and metabolites, and accuracy were independently extracted by two authors.

**Results** Among 4613 initially identified studies on metabolomics, 68 were read in full text and 32 articles were included. Studies were excluded due to duplicated data, study design or lack of identification of metabolites. Metabolomics was applied mainly in the second trimester; the most common technique was liquid-chromatography coupled to mass spectrometry. Among the 122 different metabolites found, there were 23 amino acids and 21 fatty acids. Most of the metabolites were involved with ammonia recycling; amino acid metabolism; arachidonic acid metabolism; lipid transport, metabolism and peroxidation; fatty acid metabolism; cell signalling; galactose metabolism; nucleotide sugars metabolism; lactose degradation; and glycerolipid metabolism. Only citrate was a common metabolite for prediction of early-onset and late-onset pre-eclampsia. Vitamin D was the only metabolite in common for pre-eclampsia and gestational hypertension prediction. Meta-analysis was not performed due to lack of appropriate standardised data.

**Conclusions and implications** Metabolite signatures may contribute to further insights into the pathogenesis of pre-eclampsia and support screening tests. Nevertheless, it is mandatory to validate such methods in larger studies with a heterogeneous population to ascertain the potential for their use in clinical practice.

**PROSPERO registration number** CRD42018097409.

## Strengths and limitations of this study

► This systematic review presents possible biomarkers of pre-eclampsia that could provide a basis for further research in this area.

► There was great heterogeneity among included studies due to the complexity of metabolomics procedures, which are influenced by sample collection, storage and preparation, analytical platform applied and statistical tests performed.

► Our findings must be treated with caution due to the lack of standardisation of pre-eclampsia definition (early and late-onset pre-eclampsia) and timing of sample collection in pregnancy.

## INTRODUCTION

Hypertensive disorders have been reported as associated with adverse outcomes in pregnancy since the 1970s. They are a heterogeneous group of conditions that include chronic hypertension, pre-eclampsia, gestational hypertension and pre-eclampsia superimposed on chronic hypertension.[1]

Chronic hypertension is the condition of systolic blood pressure ≥140 mm Hg and/or diastolic blood pressure ≥90 mm Hg before 20 weeks of gestation or even before pregnancy.[2] Gestational hypertension is the isolated elevation of pressure after twenty weeks of gestation. It is less damaging than pre-eclampsia but around 30% of these cases progress to pre-eclampsia.[3] The most prevalent form—pre-eclampsia—is a serious and potentially life threatening condition to the mother and the baby,[2] and its high prevalence rate persists over the years, ranging from 3% to 10% of all pregnancies, according to the country studied. Unfortunately, treatment has been the same over decades and relies on the management of clinical features, such as new onset of hypertension and end-organ dysfunction that characterises pre-eclampsia. Outcomes mostly depend on accurate diagnosis and decision on timing of

BMJ

childbirth since there are few predictive tools available.[3] As a result, one of the most urgent priorities is to identify those women at high-risk for the disease, who would be candidates for prophylactic and increased surveillance measures. The increased understanding on pathophysiology would also shed light into possible new therapeutic approaches.

Several biomarkers have been studied as predictive tools for pre-eclampsia, for example, soluble fms-like tyrosine kinase-1, soluble endoglin, markers of apoptosis and inflammation, placental protein 13, C reactive protein and markers of placental hypoxia and distress.[4] However, none are sufficiently sensitive or specific to predict pre-eclampsia in advance.

Recently, novel technologies, such as metabolomics, have been applied to predict pre-eclampsia. Metabolomics is known as the newest member of the 'omics' family. Metabolome—a collection of metabolites—is defined and used in research to pursue the phenotypic signature of a disease of interest. Metabolites are low-molecular-weight chemicals (<1500 Da) resulting from changes in gene, protein expression[5] and environmental interferences. Metabolomics methods include hydrogen nuclear MR spectroscopy ($^1$H-NMR) and gas-chromatoquid and liquid-chromatoquid-chromatography-mass spectrometry (GC and LC-MS, respectively).[5 6] Metabolomics has been applied to improve disease biomarkers and understand the pathogenesis of many conditions, such as cancer and diabetes.[7 8] One could argue its value in evaluating pre-eclampsia, which has a complex aetiology and is considered a multifactorial disease. Metabolomics, however, a high throughput technique, seems perfectly adequate, since it can simultaneously encompass a wide range of metabolic pathways. The objective of this systematic review was to determine the accuracy of metabolomics in predicting pregnancy-induced hypertensive disorders.

## METHODS

This systematic review was conducted based on our previously published protocol[9] and is reported in accordance with the Preferred Reported Items for Systematic Reviews and Meta-Analyses (PRISMA) statement[10]; a PRISMA checklist is provided as an online supplemental material 1. Our research question was: 'what is the performance of metabolomics for predicting gestational hypertensive disorders?'

### Population and interventions

Our domain of interest was any form of hypertensive disorder developed during pregnancy, in relation to either single or multiple gestations. Metabolomics was the intervention studied. Papers eligible for inclusion in our review should mention metabolites potentially differentiating hypertensive from normotensive pregnant women. Maternal samples should have been drawn during pregnancy and before diagnosis of pregnancy-induced hypertensive disorder. We excluded studies in which blood samples were collected after the diagnosis of hypertension was established, or when pre-eclampsia and gestational hypertension were analysed as the same outcome.

### Comparison and outcomes

We included studies in which women with any form of pregnancy-induced hypertension were compared with pregnant women without hypertension. Pre-eclampsia was the primary outcome, diagnosed by criteria defined by the authors.[11–15]

Secondary outcomes were: (1) early-onset pre-eclampsia, characterised by delivery <34 weeks due to pre-eclampsia[3]; (2) late-onset pre-eclampsia, requiring delivery ≥34 weeks[3]; (3) gestational hypertension, when hypertension is the only clinical finding after 20 weeks gestation; (4) pre-eclampsia superimposed on chronic hypertension; (5) white coat hypertension, that is, normal blood pressure recorded in a 24-hour ambulatory monitoring; (6) masked hypertension, characterised as normal blood pressure in the office or clinic and altered under other circumstances; and (7) transient gestational hypertension, higher blood pressure at levels during a certain period that is later normalised after repeated blood pressure readings.[11 16]

### Search strategy

Two independent researchers (JM, DFL) performed an electronic database search in PubMed, EMBASE, Scopus, Web of Knowledge, Latin American and Caribbean Health Sciences Literature, Scientific Electronic Library Online, Health Technology Assessment, and Database of Abstracts of Reviews of Effects, including studies published from 1998 to 2022. The full search strategies are provided as online supplemental material 2. We have applied no filters, except for SCOPUS (we have used the Title, Abstract and Keyword filter). The grey literature (conference abstracts) and the reference list of included studies were searched for additional articles for inclusion.

There were no language restrictions; only cohort and case control studies were included. Letters to the editor, editorials, comments, expert opinions, any type of review, experimental animal studies, case reports, intervention trials, and cross-sectional studies were all excluded. We intend to show the capacity of metabolomics to predict hypertensive disorders without interference of any intervention. Thus, it is absolutely necessary to guarantee the temporal relation between timing of sample (before) and outcome (after). This is the reason for including just cohort and case control studies.

### Data extraction and quality assessment

Trained reviewers (JM, DFL) proceeded studies' screening (using EndNote software), data extraction and synthesis, and risk of bias evaluation (Quality Assessment of Diagnostic Accuracy Studies-2[17]). Disagreements were solved by a third reviewer (MLC).

## Summary of data and meta-analysis

For all included studies, we have extracted data on research characteristics and metabolomics details. If the study comprised training and validation datasets, we have extracted metabolomics data from the validation group of women. When the study has not definitively identified a given metabolite, we have extracted all possible metabolites. We have extracted data of metabolites combined with other biomarkers, when available. We have used the Human Metabolome Database,[18] the Kyoto Encyclopaedia of Genes and Genomes[19] and the Lipid Maps[20] for matching each metabolite chemical class and subclass, and metabolic pathways and processes.

When available, data on accuracy measures on metabolomics performance in predicting hypertensive disorders of pregnancy were extracted. We planned to perform the estimation of likelihood ratios and hierarchical summary receiver operator characteristic curve,[21] and the assessment of heterogeneity and publication bias.[22]

## Patient and public involvement

Considering this is a systematic review, patients and the public were not involved in this study at all.

## RESULTS

### Main findings

The literature search for this systematic review was performed in June 2019 with a further rerun in February 2022. Figure 1 summarises the flow chart of studies' selection and final inclusion. We excluded studies (online supplemental material 3) due to study design (25 studies) or lack of identification of metabolites (5 studies). Regarding duplicated data, only the most recent and complete study was included (6 studies).

Overall, 32 studies were selected: 25 case control studies[23–47] and 7 cohort studies[48–54] (table 1). Studies from the same research project but evaluating different metabolites[43 46] or populations[24 28 29 33 36 38 39] were included. Most studies were

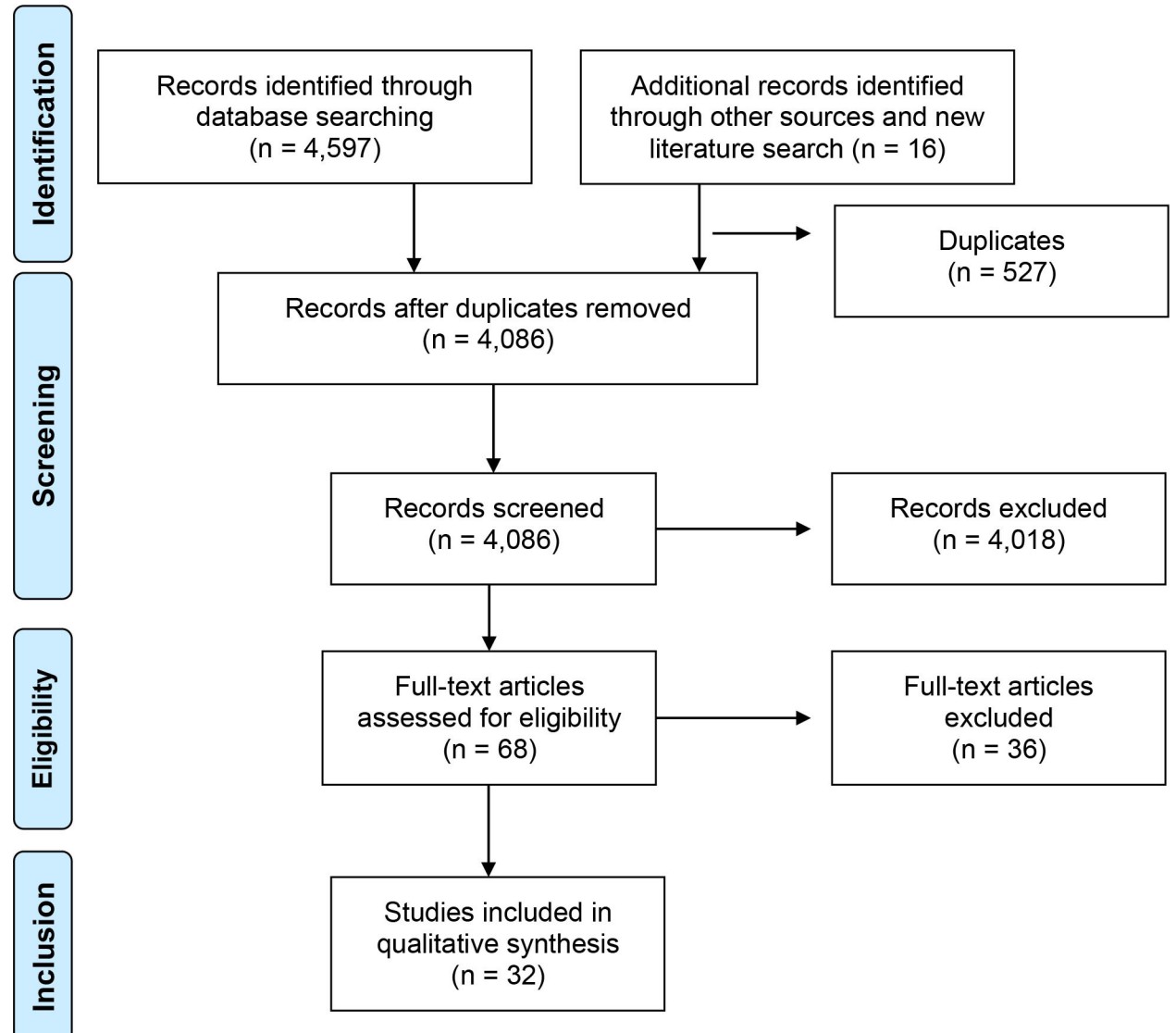

**Figure 1** Flow chart of study identification, screening, eligibility and inclusion. From Moher *et al*.[78]

**Table 1** Characteristics of included studies

| Authors, year | Country of data collection | Study design | Classification of hypertensive syndrome | Comments |
|---|---|---|---|---|
| Chappell et al 2002[23] | UK | Case–control | ISSHP, 1988 | 17 women with pre-eclampsia; 21 without pre-eclampsia. Chronic hypertensive women included. |
| Powe et al 2010[34] | USA | Nested case–control | ICD codes | 39 pre-eclampsia, 131 controls. Gestational diabetes excluded. |
| Kenny et al 2010[24] | New Zealand; Australia | Nested case–control | ASSHP, 2000 | Discovery phase (New Zealand): 60 women with pre-eclampsia, 60 matched controls (age, ethnicity, body mass index) with uneventful pregnancies. Validation phase (Australia): 39 women with pre-eclampsia, 40 healthy matched pregnancies (SCOPE study). |
| Odibo et al 2011[35] | USA | Nested case–control | ACOG, 2002 | 41 women with pre-eclampsia, 41 normotensive controls. Pregestational diabetes included. |
| Woodham et al 2011[41] | USA | Nested case–control | ACOG, 2002 | 41 women with severe pre-eclampsia, 123 normotensive uncomplicated pregnancies at term. Chronic illness and multiple pregnancies excluded. |
| Rijvers et al 2013[48] | Netherlands | Retrospective cohort | NHBP, 2000 | 06 women with pre-eclampsia, 11 with gestational hypertension, 18 normotensive women |
| Khalil et al 2013[42] | UK | Case–control | ISSHP, 2001 | 25 women with early pre-eclampsia, 50 cases of late pre-eclampsia, 300 normotensive controls (matching one case: 4 controls). Only singleton pregnancies; chronic hypertensive women included. |
| Diaz et al 2013[49] | Portugal | Prospective cohort | ACOG, 2002 | 09 women with preterm pre-eclampsia (31–37 weeks); 84 women with healthy term pregnancies |
| Kenny et al 2013[39] | New Zealand; Australia | Nested case–control | ASSHP, 2000 | 49 women with pre-eclampsia and 49 normotensive nulliparous women. (SCOPE study). |
| Kuc et al 2014[43] | Netherlands | Nested case–control | ISSHP, 2001 | 68 early-onset pre-eclampsia, 99 late-onset pre-eclampsia, 500 controls |
| Wetta et al 2014[44] | USA | Nested case–control | ACOG, 2002 | 89 women with preterm pre-eclampsia; 177 normotensive and term pregnancies |
| Bahado-Singh et al 2015[50] | UK | Prospective cohort | ISSHP, 2001 | 50 cases of early pre-eclampsia, 108 normotensive controls. Singleton pregnancies delivered at term; birth weight adequate for gestational age. HELLP syndrome cases excluded. |
| Eichelberger et al 2015[45] | USA | Nested case–control | ACOG, 2002 | 25 women with severe pre-eclampsia; 87 normotensive, healthy pregnancies delivered at term. Chronic hypertensive women and multiple pregnancies excluded. |
| Koster et al 2015[46] | Netherlands | Nested case–control | ISSHP, 2001 | 68 early-onset pre-eclampsia, 99 late-onset pre-eclampsia, 500 controls |
| Bilodeau et al 2015[54] | Canada | Prospective cohort | CHS, 1997 | 33 pre-eclampsia, 60 controls (MIROS study) |
| Ates et al 2016[51] | Turkey | Prospective cohort | ACOG, 2002 | 04 women with pre-eclampsia; 11 women with gestational hypertension; 214 normotensive women |
| Cantonwine et al 2016[47] | USA | Nested case–control | ACOG, 2002 | 50 women with pre-eclampsia, 431 without pre-eclampsia. Chronic hypertensive women included. |
| Kiely et al 2016[36] | Ireland | Nested case–control | ASSHP, 2000 | 68 pre-eclampsia, 1528 controls. SGA infants excluded from controls. (SCOPE Study) |

Continued

**Table 1** Continued

| Authors, year | Country of data collection | Study design | Classification of hypertensive syndrome | Comments |
|---|---|---|---|---|
| Bahado-Singh et al 2017[25] | UK | Prospective cohort | ISSHP, 2001 | 59 late-onset pre-eclampsia cases, 115 normotensive controls. Singleton pregnancies delivered at term; birth weight adequate for gestational age. HELLP syndrome cases excluded. |
| Bahado-Singh et al 2017[52] | UK | Nested case–control | ISSHP, 2001 | 35 term pre-eclampsia cases (≥37 w), 65 normotensive controls. Singleton pregnancies, no major malformations; controls with birth weight adequate for gestational age. |
| Dobierzewska et al 2017[26] | Chile | Case–control | ACOG, 2002 | 07 women with pre-eclampsia; 07 normotensive women |
| Ye et al 2017[27] | China | Nested case–control | ACOG, 2013 | 74 women with pre-eclampsia, 99 normotensive term pregnancies as controls. Only singleton pregnancies; chronic illness excluded in control group. |
| Gong et al 2018[28] | UK | Case–control | ACOG, 2002 | 134 term pre-eclampsia, 259 normotensive controls. Nulliparous women, singleton pregnancies (POP Study). |
| Tamblyn et al 2018[29] | Ireland | Case–control | ASSHP, 2000 | 25 women with pre-eclampsia, 25 normotensive women (SCOPE Study). |
| Sovio et al 2020[38] | UK | Case-cohort | ISSHP, 2001; ACOG, 2013 | Training and test with POP Study: 194 women with pre-eclampsia, 323 controls. Validation within BiB Study: 95 women with pre-eclampsia, 953 normotensive controls. |
| Huo et al 2020[37] | Shanghai, China | Case-cohort | ACOG, 2013 | 64 gestational hypertension, 71 pre-eclampsia. Singleton gestation only; chronic hypertensive women excluded. |
| Rylander et al 2020[31] | Sweden | Case-ontrol | ICD codes | 296 pre-eclampsia, 580 controls. Excluded pregnancies with SGA infants. |
| Lee et al 2020[30] | South Korea | Case–control | ACOG, 2013 | 33 pre-eclampsia, 66 controls. Singleton pregnancies, congenital malformations excluded. |
| Kenny et al 2020[33] | Ireland and England | Nested case–control | ISSHP, 2018 | 97 pre-eclampsia (23 preterm pre-eclampsia and 74 term pre-eclampsia); 335 controls. Nulliparous women, singleton pregnancies. Chronic illness and fetal malformations excluded (SCOPE study) |
| Shanmugalingam et al 2020[53] | Australia | Prospective cohort | ISSHP, 2018 | 21 pre-eclampsia, 103 controls. High-risk women. |
| Harville et al 2021[32] | USA | Case–control | Not mentioned | 18 pre-eclampsia, 109 controls. Only singleton pregnancies. |
| Huang et al 2021[32] | USA | Case–control | ACOG, 2013 | 20 pre-eclampsia, 20 controls (validation phase). |

ACOG, American College of Obstetricians and Gynecologists; ASSHP, Australasian Society of the Study of Hypertension in Pregnancy; CHS, Canadian Hypertensive Society; ICD, International Classification of Diseases; ISSHP, International Society for the Study of Hypertension in Pregnancy; MIROS, Maternal and Infant Research on Oxidative Stress; POP, Pregnancy Outcome Prediction; PTB, preterm birth; SCOPE, Screening for Pregnancy Endpoints; SGA, small for gestational age.

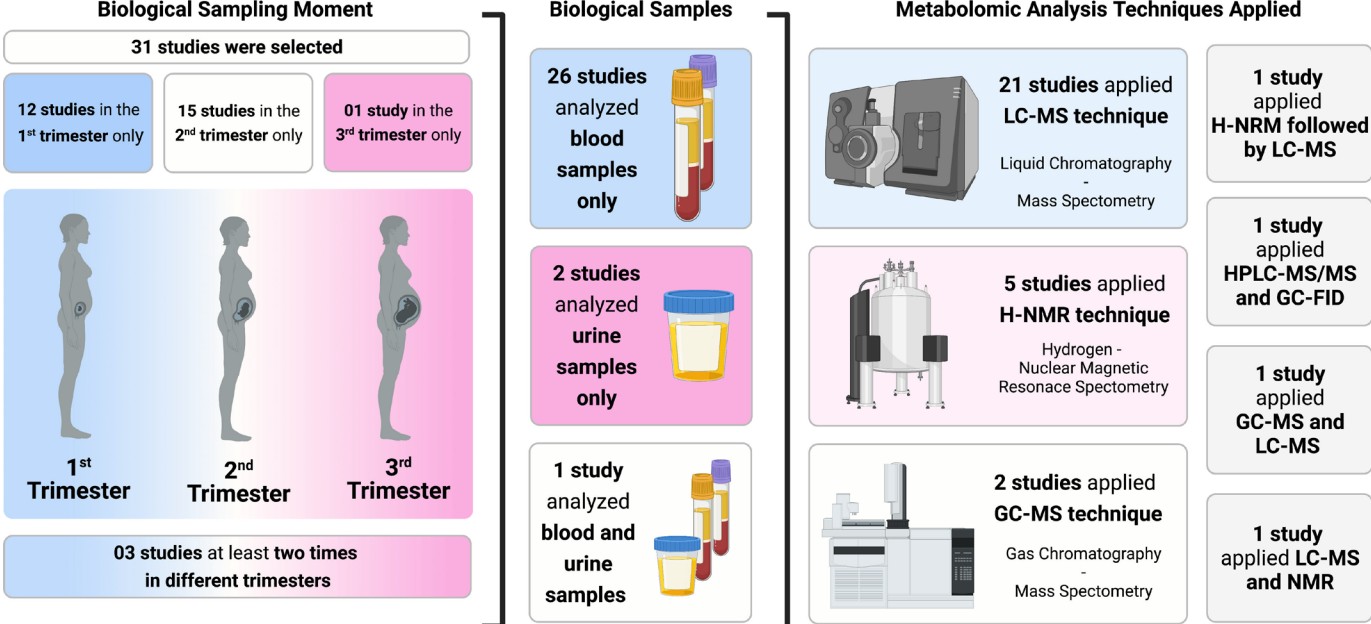

**Figure 2** Characterisation of study selection regarding biological samples, sampling moment and metabolomic analysis techniques applied. GC-FID, gas-chromatography flame ionisation detector; H-NRM, hydrogen nuclear MR spectroscopy; HPLC, high-performance liquid chromatography; LC-MS, liquid-chromatography-mass spectrometry.

developed in Europe[23 25 28 29 31 33 36 38 39 42 43 46 48–52] and North America[32 34 35 40 41 44 45 47 54]; there was only one research from Latin America,[26] two from Oceania[24 53] and another three from Asia.[27 30 37] Twelve studies have sampled maternal specimens in the first trimester,[31 32 34 35 42 43 46 48 50–53] 15 studies have sampled in the second trimester,[23 24 27 29 30 33 36–39 41 44 45 49 54] four studies have sampled at least two times[25 26 40 47] in pregnancy, and one study included women in late third trimester[28] (figure 2). Four studies[24 33 38 40] have performed training and validation with different populations.

Studies have defined pre-eclampsia as hypertension developed in pregnancy associated with proteinuria[11 13–15] or end-organ dysfunction.[12] Although not consistentlly, chronic hypertension or other illnesses,[24 27 29 33 36–39 41 45 55] and multiple pregnancies[25 27 30 32 33 36 37 39 41 42 45 52] were reasons for participants exclusion. When chronic hypertensive women were included, there were women taking aspirin as prophylaxis.[23 53] Then, there are no specific data on superimposed pre-eclampsia or multiple pregnancies. Similarly, there are no studies on white coat hypertension, masked hypertension, or transient gestational hypertension. Only three studies have brought data on gestational hypertension.[32 37 51] Small for gestational age (SGA) infants were fully excluded[31] or included only in pre-eclampsia group, either because they were excluded from control groups,[23 34] or because this variable was not used as for matching cases and controls.[24 29 36]

In table 2, characteristics of the included studies regarding metabolomics techniques and accuracy measures are shown according to the primary and secondary outcomes. The majority of studies did not present data according to gestational age of delivery. The most common technique was LC-MS,[24 26 27 29 31 33–38 40 41 43–48 51 53 55] followed

by H-NMR[25 39 49 50 52] and GC-MS.[23 42] Bahado-Singh et al[25] (H-NMR followed by LC-MS), Bilodeau et al[54] (high-performance liquid chromatography-MS/MS and GC-flame ionisation detector), Lee et al[30] (GC-MS and LC-MS), and Harville et al[32] (LC-MS and NMR) have used two complementary techniques. Most studies analysed blood samples,[23 24 26–28 30–46 48 50–54] two studies analysed urine[47 49] and one study analysed both, urine and blood[29] (figure 2). There were no studies carrying out analyses on maternal hair or amniotic fluid. All studies that used H-NMR[25 32 39 49 50 52] and four that applied LC-MS[24 28 32 38] were untargeted evaluations. Eleven studies reported lipidomic analysis and/or amino acid analysis.[23 26 30 35 40 42 43 46 48 53 54] Six studies described vitamin D analysis, all of them using LC-MS.[29 34 36 41 44 51] Five studies reported environmental exposures, focusing on caffeine,[45] bisphenol A,[27] phthalates[47] and per and polyfluoroalkyl substances (PFAS).[31 37] Two studies have looked for differences in pre-eclampsia prediction according to foetal sex.[28 47] There were five metabolites in common between pre-eclampsia and SGA prediction (sphinganine 1-phosphate, sphingosine 1-phosphate, tyrosine, vitamin D, N1, N12-diacetylspermine).[56]

For pre-eclampsia requiring delivery at any gestational age, the lowest area under the curve (AUC) was for isoprostane 8-epi-prostaglandin F2α (0.55),[23] and the highest (AUC 0.95) was for a model composed by cerasinone, bolasterone, and two unidentified metabolites, presented as retention time and mass (6.30_477.7721 m/z, 5.86_313.1070 m/z).[32] For early-onset pre-eclampsia, the AUC of untargeted metabolomics in first trimester maternal blood varied from 0.673 (ultraperformance liquid chromatography-MS, UPLC-MS/MS)[38] to 0.835

**Table 2** Metabolites and their accuracy measures for predicting pre-eclampsia according to gestational age of required delivery

| Authors, year | Metabolomics methods | Maternal biological specimen; gestational age at sampling | Increased metabolites | Decreased metabolites | S/Sp/ AUC (95% CI) |
|---|---|---|---|---|---|
| **Pre-eclampsia at any gestational age** | | | | | |
| Chappell et al 2002[23] | GC-MS | Blood; 20 weeks | Isoprostane 8-epi-prostaglandin $F_{2\alpha}$ | | -/-/ 0.62 (0.44 to 0.81) |
| | | Blood; 24 weeks | Isoprostane 8-epi-prostaglandin $F_{2\alpha}$ | | -/-/ 0.55 (0.35 to 0.75) |
| Powe et al 2010[34] | Targeted LC-MS/MS | Blood (serum); 11 weeks (±3weeks) | | 25(OH)vitamin D* | -/-/- |
| Odibo et al 2011[35] | Targeted LC-MS/MS | Blood (serum); 11–14 weeks | Hydroxyhexanoylcarnitine, phenylalanine, glutamate, alanine | | -/-/ 0.82 (0.80 to 0.85) |
| Woodham et al 2011[5] | Targeted LC-MS/MS | Blood (serum); 15–20 weeks | | 25(OH)vitamin D | -/-/ 0.745 |
| Diaz et al 2013[49] | Untargeted $^1$H-NMR | Urine; 14–26 weeks | N-methyl-2-pyridone-5-carboxamide, 3-methylhistidine, 4-deoxyerythronic acid, cis-aconitate, citrate, Glutamine, hippurate, indoxyl sulphate, N-methylnicotinamide, sucrose, U35 | 2-ketoglutarate, 4-OH-hippurate, acetate, carnitine, creatinine, formate, fumarate, galactose, isoleucine, lactose, phenylacetylglutamine, p-cresol sulphate, scyllo-ino, succinate, trigonelline, tyrosine, U14, U4, U10 | -/-/- |
| Kenny et al 2013[39] | Untargeted H-NMR | Blood (serum); 15weeks | | Acetamide, Glutamine, Trimethylamine, creatinine, dimethyl sulfone, valine, isoleucine, ornithine, glycine, methionine, betaine, acetate, proline | -/-/- |
| Rijvers et al 2013[48] | Targeted UPLC-MS/MS | Blood (plasma); 12weeks | L-arginine/ Asymmetric dimethylarginine ratio | | -/-/- |
| | | Blood (plasma); 16–20 weeks | | L-arginine | -/-/- |
| Wetta et al 2014[44] | Targeted LC-MS/MS | Blood (serum); 15–21 weeks | | 25(OH)vitamin D* | -/-/- |
| Bilodeau et al 2015[54] | Targeted HPLC-MS/MS | Blood (plasma); 12–18 weeks | (+/-) 5-iPF2α-VI, (+/-) 5-iPF2α-VI-(d11) | | -/-/ 0.67 |
| | GC-FID | Blood (plasma); 12–18 weeks | | Alfa-linolenic acid, stearidonic acid, eicosatrienoic acid, eicosatetraenoic acid, eicosapentaenoic acid, docosapentaenoic acid, docosahexaenoic acid | -/-/- |
| Ates et al 2016[51] | Targeted LC-MS/MS | Blood (serum); 11–14 weeks | | Vitamin D (<10mg/dL) | 0.50/0.54 |
| Cantonwine et al 2016[47] | Targeted HPLC-MS | Urine; 4–16 weeks | Total bisphenol A, Σ di(2-ethylhexyl) phthalate, mono-ethyl phthalate | %mono(2-ethylhexyl) phthalate | -/-/- |
| | | Urine; 22–29 weeks | mono(2-ethylhexyl) phthalate, Σ di(2-ethylhexyl) phthalate, mono(3-carboxypropyl) phthalate | monoisobutyl phthalate | -/-/- |
| | | Urine; 33–38 weeks | mono(2-ethylhexyl) phthalate, Σ di(2-ethylhexyl) phthalate, mono(3-carboxypropyl) phthalate | | -/-/- |
| Kiely et al 2016[36] | Targeted LC-MS/MS | Blood (serum); 14–16 weeks | | 25(OH)D3, 3-epi-25(OH)D3, and 25(OH)D2* | -/-/- |

Continued

**Table 2** Continued

| Authors, year | Metabolomics methods | Maternal biological specimen; gestational age at sampling | Increased metabolites | Decreased metabolites | S/Sp/ AUC (95% CI) |
|---|---|---|---|---|---|
| Dobierzewska et al 2017[26] | Targeted HPLC-MS | Blood (plasma); 11–14 weeks | | Sphingomyelin 16:0, sphingomyelin 18:0, ceramide 14:0 | -/-/ - |
| | | Blood (plasma); 32–36 weeks | | Ceramide 14:0, ceramide 24:0 | -/-/ - |
| Ye et al 2017[27] | Targeted LC-MS | Blood (serum); 16–20 weeks | Free bisphenol A | | -/-/ 0.73 (0.65 to 0.81) |
| Tamblyn et al 2018[29] | Targeted LC-MS/MS | Urine; 14–16 weeks | | 25(OH) vitamin $D_3$; 24,25(OH)two vitamin D3* | -/-/ - |
| Huo et al 2020[37] | HPLC/MS-MS | Blood (serum); 13–17 weeks | | Perfluorooctanate; perfluorooctane sulfonate; perfluorodecanoic acid; perfluoroundecanoic acid; perfluorononanoic acid; perfluorohexanesulfonate; perfluoroheptanoic acid; perfluorobutane sulfonate; perfluorododecanoic acid; perfluorooctane sulfonamide* | -/-/ - |
| Rylander et al 2020[31] | Targeted LC-MS/MS | Blood (serum); 12–14 weeks | Perfluorohexane sulfonate (0.53–0.78 ng/mL) | | 0.38/0.67/- |
| Lee et al 2020[30] | GC-TOF MS | Blood (plasma); 16–24 weeks | | Propane-1,3-diol | 0.75/0.83/0.868 (0.844 to 0.891) |
| | LC-Orbitrap MS | Blood (plasma); 16–24 weeks | LysoPE C20:0; SM C30:1 | LysoPC C19:0; SM C28:1 | |
| Kenny et al 2020[33] | Targeted LC-MS/MS | Blood (plasma); 14–16 weeks | Dilinoleoyl-glycerol (DLG) | | 0.39/0.90/0.70 (0.59 to 0.82) |
| Shanmugalingam et al 2020[53] | Targeted LC-MS/MS | Blood (plasma); 12 weeks | | 15-epilipoxin-A4 | |
| Harville et al 2021[32] | Untargeted UPLC-HR-MS | Blood (serum); 6–14 weeks | Cerasinone; bolasterone, 6.30_477.7721 m/z, 5.86_313.1070 m/z† | | -/-/ 0.95 |
| | NMR | Blood (serum); 6–14 weeks | | Asparagine, N,N-Dimethylglycine, Trimethylamine | -/-/ - |

**Table 2** Continued

| Study | Method | Sample | Metabolites | | Value |
|---|---|---|---|---|---|
| Huang et al 2021[40] | LC-MS | Blood (serum); 5–15 weeks | | Cer(d18:1/25:0) | -/-/0.868 |
| | | Blood (serum); 16–29 weeks | | Cer(d18:1/25:0) | -/-/0.747 |
| Early onset pre-eclampsia (<34 w) | | | | | |
| Odibo et al 2011[35] | Targeted LC-MS/MS | Blood (serum); 11–14 weeks | Hydroxyhexanoylcarnitine, phenylalanine, glutamate, alanine | | -/-/0.84 |
| Khalil et al 2013[42] | Targeted GC-MS; GC-MS/MS | Blood (plasma); 11–13 weeks | asymmetric dimethylarginine/ L-arginine ratio; asymmetric dimethylarginine/ L-homoarginine ratio | L-arginine, L-homoarginine | -/-/- |
| Kuc et al 2014[43] | Targeted UPLC-MS/MS | Blood (serum); 8–13 weeks | | Taurine, asparagine | -/-/- |
| Bahado-Singh et al 2015[50] | Untargeted 1H-NMR | Blood (serum); 11–13 weeks | 2-hydroxybutyrate, 3-hydroxyisovalerate, citrate | Acetone, glycerol | 0.75/0.74/0.835 (0.769 to 0.941) |
| Eichelberger et al 2015[45] | LC-MS | Blood (serum); 16–19 weeks | | Paraxanthine/caffeine ratio | -/-/- |
| Koster et al 2015[46] | Targeted UPLC-MS/MS | Blood (serum); 8–13 weeks | Hexanoylcarnitine, octenoylcarnitine, octanoylcarnitine, decenoylcarnitine, decanoylcarnitine, dodecanoylcarnitine, lauroylcarnitine, tetradecenoylcarnitine, hexadecenoylcarnitine | Isobutyrylcarnitine, linoleylcarnitine, stearoylcarnitine | -/-/- |
| Sovio et al 2020[38] | Untargeted UPLC-MS/MS | Blood (serum); 12 weeks / Blood (serum); 20 weeks / Blood (serum); 28 weeks | 4-hydroxyglutamate | | ▲ /-/0.673 ▲ /-/0.731 ▲ /-/0.765 |
| Late onset pre-eclampsia (≥34 w) | | | | | |

Continued

**Table 2** Continued

| Study | Method | Sample; gestational age | Metabolites | | Value (-/-/-) |
|---|---|---|---|---|---|
| Kenny et al 2010[24] | Untargeted UPLC-MS | Blood (plasma); 14–16 weeks | Monosaccharide(s), decanoylcarnitine, oleic acid, docosahexaenoic acid and/or docosatriynoic acid, gama-butyrolactone and/or oxolan-3-one, 2-oxovaleric acid and/or oxo-methylbutanoic acid, acetoacetic acid, hexadecenoyleicosatetraenoyl-Sn-glycerol, Di-(octadecadienoyl)-sn-glycerol, sphingosine 1-phosphate, sphinganine 1-phosphate, vitamin $D_3$ derivatives | 5-Hydroxytryptophan, methylglutaric acid and/or adipic acid | -/-/ 0.92‡ |
| Kuc et al 2014[43] | Targeted UPLC-MS/MS | Blood (serum); 8–13 weeks | | Glycylglycine | -/-/ - |
| Koster et al 2015[46] | Targeted UPLC-MS/MS | Blood (serum); 8–13 weeks | Hexanoylcarnitine, octenoylcarnitine, octanoylcarnitine, decenoylcarnitine, decanoylcarnitine, dodecanoylcarnitine, tetradecenoylcarnitine | Linoleylcarnitine, palmitoylcarnitine, oleylcarnitine, stearoylcarnitine | -/-/ - |
| Cantonwine et al 2016[17] | Targeted HPLC-MS | Urine; 4–16 weeks | mono(2- ethyl-5-oxohexyl) phthalate, mono(2-ethyl-5-carboxypentyl) phthalate, Σ di(2-ethylhexyl) phthalate, mono-ethyl phthalate | | -/-/ - |
| | | Urine; 33–38 weeks | mono(2- ethyl-5-oxohexyl) phthalate, mono(2-ethyl-5-carboxypentyl) phthalate, Σ di(2-ethylhexyl) phthalate, Mono-*n*-butyl phthalate | | -/-/ - |
| Bahado-Singh et al 2017[25] | Untargeted $^1$H-NMR | Blood (serum); 11–13 weeks | Carnitine, pyruvate, acetone | – | 0.30/0.80/0.629 (0.490 to 0.767) |
| Bahado-Singh et al 2017b[52] | Untargeted $^1$H-NMR, targeted LC-MS/MS | Blood (serum); 11–13 weeks | – | Putrescine (LC-MS); urea, carnitine (H-NMR) | 0.72/0.57/0.701 |
| | | Blood (serum); 32–33 weeks | Methylhistidine, propylene glycol (H-NMR), citrate, hexose§ | Serotonin (LC-MS) | 0.74/0.72/0.761 |
| Gong et al 2018[28] | Untargeted LC-MS/MS | Blood (serum); 36 weeks | N1, N12-diacetylspermine¶ | – | – |
| **Gestational hypertension** | | | | | |
| Ates et al 2016[51] | Targeted LC-MS/MS | Blood (serum); 11–14 weeks | | Vitamin D (<10 mg/dL) | 0.45/0.54 |
| Huo et al 2020[37] | HPLC/ MS-MS | Blood (serum); 13–17 weeks | Perfluoroheptanoic acid** | | -/-/ - |

Continued

**Table 2** Continued

| Harville et al 2021[32] | Untargeted UPLC-HR-MS | Blood (serum); 6–14 weeks | Neomenthol-glucuronide, dibenzylamine, pilocarpine, 2,6-Di-tert-butyl-4-hydroxymethylphenol, 0.59_746.6045n, 8.66_762.1452m/z, 12.74_412.2842m/z | -/ - / 0.95† |

*No statistical difference between pre-eclampsia and normotensive controls.

†AUC for all metabolites model; some of them were not identified, thus they are presented as retention time and mass; it is unclear which metabolite was up or downregulated in pre-eclampsia cases.

‡Accuracy measures for the validation phase.

§Unclear type of platform that has found this metabolite, or if it is increased or decreased in pre-eclampsia.

¶Higher pre-eclampsia risk with increasing N1, N12-diacetylspermine concentrations.

**No statistical difference between pre-eclampsia normotensive controls.

AUC, area under the curve; GC-FID, gas-chromatography flame ionisation detector; GC-MS, gas-chromatography coupled to mass spectrometry; GC-TOF, GC-time-of-flight; ¹H-NMR, hydrogen nuclear MR; HPLC-MS, high performance liquid chromatography-MS; LC-MS, liquid-chromatography coupled to MS; LC-Orbitrap MS, liquid chromatography Orbitrap MS; S, sensitivity; Sp, specificity; UPLC-MS, ultraperformance liquid chromatography-MS.

(H-NMR).[50] For late-onset pre-eclampsia, the AUC of untargeted metabolomics in first trimester maternal blood was 0.629 (H-NMR)[52]; in second trimester, 0.92 (UPLC-MS)[24] and in third trimester, 0.761 (H-NMR and LC-MS/MS).[25]

When analysed together with non-metabolomics biomarkers (table 3), the best performance was achieved by taurine combined with maternal prior risk and mean arterial blood pressure (MAP) (AUC 0.93 in first trimester for early onset pre-eclampsia)[43]; followed by 3-hydroxyisovalerate, arginine, glycerol combined with uterine arteries pulsatility index (AUC 0.917 in first trimester for early onset pre-eclampsia)[50]; and leptin/ceramide(d18:1/25:0) ratio in in first and second trimester (AUC 0.876 and 0.892, respectively) for pre-eclampsia at any gestational age.[40]

All predictive metabolites of hypertensive disorders of pregnancy according to chemical class and subclass, and metabolic pathways possibly involved are presented as online supplemental material 4. Ten metabolites could not be found in the metabolome databases (U4, U10, U14, U35,[49] hexadecenoyleicosatetraenoyl-Sn-glycerol[24]; SM C28:1, SM C30:1, LysoPC C19:0,[30] dibenzylamine,[32] perfluoroheptanoic acid[37]). There were 121 different metabolites, from ten different super classes. The majority of metabolites corresponded to amino acids, peptides, and analogues (23 compounds); fatty acid esters (11 compounds); fatty acids and conjugates (10 compounds); and carbohydrates and carbohydrate conjugates (5 compounds). They are mainly involved with ammonia recycling; amino acid metabolism; arachidonic acid metabolism; lipid transport, metabolism and peroxidation; fatty acid metabolism; cell signalling; galactose metabolism; nucleotide sugars metabolism; lactose degradation; and glycerolipid metabolism. There were 10 benzoic acids and derivatives, but only hippurate has a known pathway. Citrate was found in common for prediction of pre-eclampsia in any gestational age,[49] early[50] and late-onset[25] disease. Octenoylcarnitine,[46] octanoylcarnitine,[46] acetone,[25 50] linoleylcarnitine,[46] stearoylcarnitine,[46] tetradecenoylcarnitine[46] were predictive for both early and late pre-eclampsia. Vitamin D was the only metabolite involved simultaneously with pre-eclampsia,[41] late onset[24] pre-eclampsia and gestational hypertension.[51]

## Summary of included studies

The untargeted studies provided a great number of predictive metabolites. Kenny et al showed a decrease of 13 metabolites (NMR)[39] and an increment of another 40 metabolites (LC-MS)[24] in blood analysis between 14 and 16 weeks of nulliparous pregnant women exclusively. In a validation phase with a list of 14 different metabolites, the OR of developing pre-eclampsia, at any gestational age, was 23 (95% CI 7 to 73), with AUC 0.92.[24] These same metabolites were validated in a different population,[33] and the dilinoleoyl-glycerol has achieved an AUC 0.70 (95% CI 0.59 to 0.82).

**Table 3** Accuracy measures of metabolites and other pregnancy-related biomarkers

| Maternal specimen | Trimester of pregnancy | Metabolites | Additional variables considered in prediction models | AUC of metabolites+additional variables model (95% CI) |
|---|---|---|---|---|
| Pre-eclampsia at any gestational age | | | | |
| Blood | First | Dilinoleoyl-glycerol heptadecanoyl-2-hydroxy-sn-glycero-3-phosphocholine | PlGF | 0.78 (0.69 to 0.88)[29] |
| | | Ceramide(d18:1/25:0)* | Leptin | 0.876[32] |
| Blood | Second | 25OHD | VEGF +sFLT-1/PlGF | 0.851[5] |
| | | Ceramide(d18:1/25:0)* | Leptin | 0.892[32] |
| Blood | Third | 4-hydroxyglutamate | s-Flt-1/PlGF | 0.834[25] |
| Early onset pre-eclampsia (delivery <34 weeks) | | | | |
| Blood | First | Taurine | Prior risk +MAP | 0.93[10] |
| | | Stearoylcarnitine | Prior risk +MAP | 0.747[14] |
| | | 3-hydroxyisovalerate, arginine, glycerol | UtPI | 0.917[12] |
| Late onset pre-eclampsia (delivery ≥34 weeks) | | | | |
| Blood | first | Pyruvate, carnitine, glycerol | UtPI | 0.722[19] |
| Blood | first | Stearoylcarnitine | Prior risk +MAP | 0.828[14] |
| Blood | first and third | Urea, SM C18:1 (first), Hexose, Citrate (third) | Maternal BMI (12 w)+MAP (32 w) | 0.805[20] |

*Increased leptin/Ceramide(d18:1/25:0) ratio in pre-eclamptic women.
AUC, area under the curve; BMI, body mass index; MAP, mean arterial pressure; UtPI, uterine arteries pulsatility index.

Bahado-Singh *et al* applied metabolomics to predict early-onset and late-onset cases.[25 50 52] Using maternal first trimester blood (11–13 weeks), those authors reinforced the theory of the multiple phenotypes for pre-eclampsia.[57 58] 2-hydroxybutyrate was identified as predictor of early-onset pre-eclampsia,[50] whereas carnitine, pyruvate and acetone resulted in an AUC of 0.629 for late-onset disease. This latter AUC increased to 0.734 by combining metabolites and uterine Doppler velocimetry and maternal weight.[52] Finally, in a two-step evaluation in the first and third trimester to predict term pre-eclampsia (≥37 weeks), a combination of metabolomics and proteomics yielded an AUC of 0.817.[25]

Through an untargeted urinary metabolome analysis, Diaz *et al*[49] showed decreased levels of acetate, formate, fumarate (all involved with the Krebs cycle), succinate and isoleucine; and increased levels of 4-deoxyerythronic acid, a degradation product of 3-hydroxybutyrate[49] between 14 and 26 weeks. Finally, still working with an untargeted approach, Sovio *et al* demonstrated an increase of 4-hydroxyglutamate serum levels among women who developed early-onset pre-eclampsia, measured at 12 weeks.[38] Its prediction performance was improved by addition of pregnancy-associated plasma protein A and PlGF.[38]

Regarding lipidomic or amino-acid studies, Odibo *et al* adjusted four metabolites (hydroxyhexanoylcarnitine, phenylalanine, glutamate, alanine) for maternal variables (maternal body mass index, ethnicity and diabetes) and the AUCs were similar with the four-metabolite model (0.82) or when only the three amino acids were used for predicting pre-eclampsia (0.81). Additionally,

the four metabolites have demonstrated equal AUCs for predicting pre-eclampsia at any gestational age (0.82) or in cases of early onset disease (0.84).[35] In the first trimester, Shanmugalingam *et al*[53] showed lower levels of 15-epilipoxin-A4, which is possibly involved with the aspirin mechanism of pre-eclampsia prophylaxis. Chappell *et al* showed an increment in 8-epi-prostaglandin F2α from 20 to 24 weeks in the pre-eclampsia group.[23] Recently, Bilodeau *et al* reinforced that F2-isoprostanes isomers from class VI were increased in the pre-eclampsia group.[54] The prostaglandins' metabolites are markers of lipid peroxidation, which pointed to an oxidative stress status, but presented modest accuracy.[23 54]

Rijvers *et al*[48] and Khalil *et al*[42] studied the metabolism of asymmetric dimethylarginine (ADMA), which is considered an important inhibitor of nitric oxide action. Rijvers *et al* found an increase of L-arginine/ADMA ratio at 12 weeks in the pre-eclampsia group,[48] and a consistent decrease in L-arginine between 16 week and 20 week in the pre-eclampsia compared with the gestational hypertension group. This can be clinically relevant since L-arginine supplementation is possible.[59] Khalil *et al* found lower levels of L-arginine and L-homoarginine at 11–13 w in the early-onset pre-eclampsia, and unchanged levels in the late-onset group, even after the excluding chronic hypertensive women.[42]

Koster *et al*[46] showed that stearoylcarnitine was a common biomarker for both early-onset and late-onset pre-eclampsia. In a model composed of stearoylcarnitine, prior risk and MAP, the detection rate improved by 45% and 21%, for early and late-onset cases of pre-eclampsia, respectively. Dobierzewska *et al*[26] have studied

sphingolipids that have been related to trophoblast differentiation and invasion. In the ceramide profile, the Cer 14:0 was decreased in both first and third trimesters of pre-eclampsia group, while Cer 16:0 was significantly increased in cross-gestational pre-eclampsia plasma samples between first and second trimesters. Plasma levels of dihydro-sphingosine-1 decreased from the first to the second trimester among the pre-eclampsia group, and SM 18:0 increased through gestation of these participants.[26] Still studying ceramides, Huang et al[40] have found decreasing levels of ceramide (d18:1/25:0) from 5 to 29 weeks, which increased the leptin/ceramide (d18:1/25:0) ratio. Lee et al proposed a model constituted by SM C28:1, SM C30:1, LysoPC C19:0, LysoPE C20:0 and propane-1,3-diol, achieving an AUC of 0.868 for identifying pre-eclampsia.[30]

Gong et al[28] analysed serum samples in different moments of pregnancy, but it was at 36 weeks when N1, N12 diacetylspermine were higher in women who later present pre-eclampsia, especially in the case of a female fetus.[28] This was the only study in common with our previous systematic review about SGA infant's prediction using metabolomics.[56]

Regarding targeted vitamin D analysis, all studies have applied LC-MS.[29 34 36 41 44 51] In general, they found lower serum vitamin D concentrations in women who later developed pre-eclampsia, both in blood[36] and urine,[29] suggesting that dysregulation of vitamin D metabolism occurs early in pregnancy. Indeed, Woodham et al have pointed reduced odds of 38% of developing severe pre-eclampsia for a 10 nmol/L increase in 25(OH)D level between 15 and 20 weeks.[41] However, results did not reach statistical significance when a threshold was applied. In nested case–controls[34 44] or cohort[51] studies, there were no differences between vitamin D status among pre-eclampsia group and controls. In the model proposed by Woodham et al combining 25OHD, VEGF and sFLT-1/PlGF ratio, the AUC was 0.851, a higher value than either marker alone.[41] Vitamin $D_3$ derivatives were found in the untargeted analysis of Kenny et al[24] but upregulated in pre-eclampsia cases.[41]

Two studies have evaluated maternal exposure to bisphenol A and its metabolites.[27 47] The di(2-ethylhexyl) phthalate was increased in all trimesters.[47] Its HR for development of pre-eclampsia varied from 1.79 (95% CI 1.30 to 1.52), with 4–16 weeks, to 2.92 (95% CI 1.61 to 5.28) with 33–38 weeks. The mono(2-ethyl-5-oxohexyl) phthalate, the mono(2-ethyl-5-carboxypentyl) phthalate and the Σdi(2-ethylhexyl) phthalate were associated to increased HRs for pre-eclampsia when evaluated in urine[47] from both the first and the second half of pregnancy. When the findings were analysed according to fetal sex, exposure to bisphenol and di(2-ethylhexyl) phthalate was only associated to pre-eclampsia in pregnancies of female fetuses.[47] In maternal serum,[27] free bisphenol was higher in pre-eclamptic women compared with normotensive controls, even when superimposed pre-eclampsia cases were excluded. The cut-off of >4.4 µg/L was associated

with increased odds of 16.46 (95% CI 5.42 to 49.95) of developing pre-eclampsia (adjusted for maternal age, parity, and body mass index).[27]

In another analysis of environmental factors, Eichelberger et al found no differences in paraxanthine or caffeine levels, but the paraxanthine/caffeine ratio was lower in pre-eclampsia (0.23 vs; normotensive women, 0.37, p=0.02).[45] This represented a decreased risk of developing pre-eclampsia with increasing paraxanthine/caffeine ratio (OR 0.53, 95% CI 0.31 to 0.90). Finally, regarding the PFASs, Rylander et al[31] and Huo et al[37] have found no differences between pre-eclamptic women and normotensive controls in blood samples from the first or second trimester, respectively. However, levels of perfluorohexane sulfonate between 0.53 and 0.78 ng/mL were associated to higher odds of pre-eclampsia (aOR 1.67, 95% CI 1.02 to 2.74)[31]; and of perfluoroheptanoic acid, to higher odds of gestational hypertension (aOR 1.38; 95% CI 1.01 to 1.87).

In the last study to evaluate gestational hypertension, Harville et al[32] identified higher concentrations of 2,6-Di-tert-butyl-4-hydroxymethylphenol in blood samples between 6 and 14 weeks. This metabolite is derived from 2,6-Di-tert-butyl-4-methylphenol, a synthetic phenolic antioxidant used widely in foods, polymers, and cosmetics to slow oxidation. It has been linked to induce cellular DNA damage.

### Quality assessment and risk of bias

The synthesised data for all included studies is shown on figure 3. Regarding the risk of bias, the majority of studies were rated low for 'Patient Selection' and 'Flow and Timing' domains. On the contrary, most studies were classified as unknown risk of bias in 'Index Test' and 'Reference Standard' domains due to missing information whether diagnosis and metabolomics analysis were interpreted independently of each other. Virtually, all studies were ranked as low with regard to applicability, with minor exceptions[32 47 49 53 60] (online supplemental material 5).

### Meta-analysis

We found that one-third of included studies were designed around a prediction aim[23–25 27 35 38 41 50 52] and we extracted accuracy measures when possible (tables 2 and 3). The other studies only presented statistical differences between groups. Unfortunately, there were discrepancies regarding type of maternal sample or gestational age at sampling, and metabolomics technique. Then, it was not possible to perform a meta-analysis.

### DISCUSSION

Hypertensive disorders constitute one of the leading causes of maternal morbidity and mortality worldwide. In this systematic review, we have found that amino acids and fatty acids were the most common chemical subclasses associated with pre-eclampsia. Similarly, the

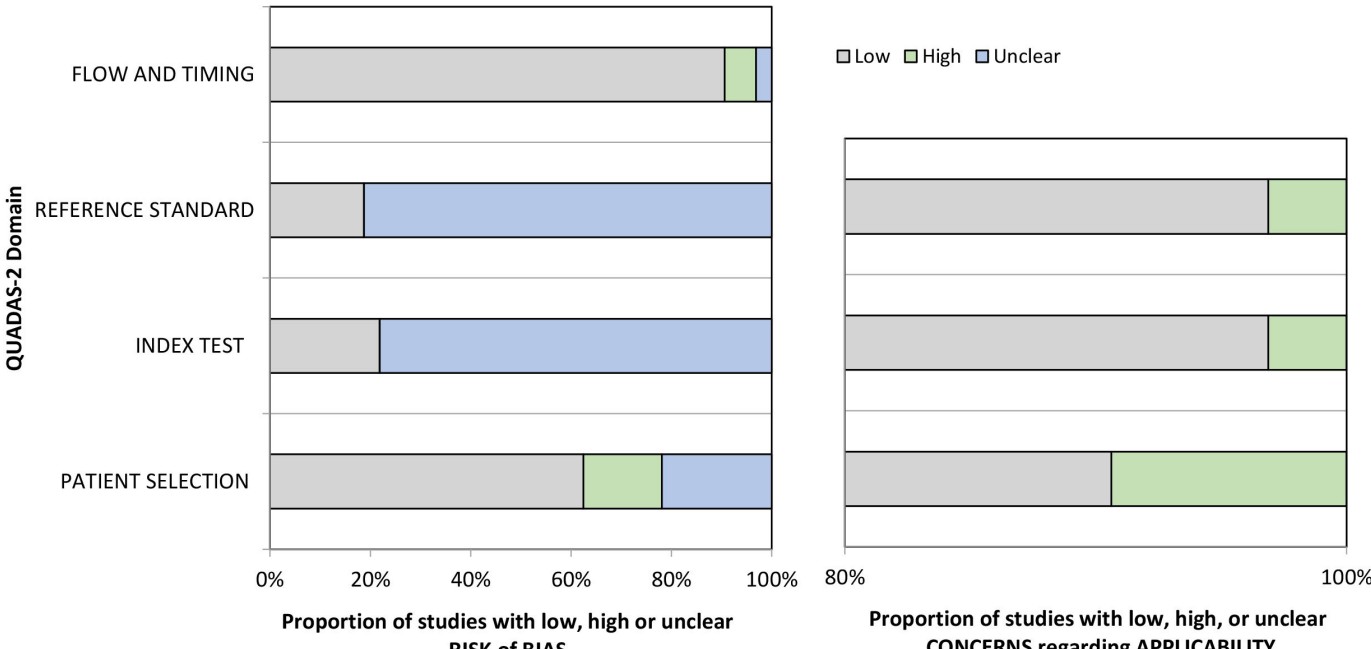

**Figure 3** Assessment of risk of bias (left) and applicability concerns (right) of included studies.

main pathways involved in pre-eclampsia were lipid metabolism, mitochondrial beta-oxidation of short chain saturated fatty acids, and cell signalling (figure 4). Our findings can improve disease understanding and contribute to future studies as a guide of pathways to be explored with the purpose to better treat the disease.

### Amino acids and related compounds

Amino acids, peptides and analogues were the most common chemical subclass. Taurine, glycine and glycylglycine were decreased in women who developed pre-eclampsia.[39 43] The function of taurine has been associated with cytoprotection, involving the placentation process and the spiral artery remodelling. Nevertheless, a reduction in taurine activity may be associated with impaired placentation, which is correlated with the pathophysiology of pre-eclampsia.[43] Glycylglycine is strongly associated with homocysteine levels and participates in the metabolic pathway that antagonises homocysteine: lower glycine levels promote higher homocysteine levels. Homocysteine is linked to cardiovascular risk and endothelial dysfunction.

Indeed, a reduction in arginine and homoarginine was observed among pre-eclampsia patients.[42] These are substrates for oxide nitric synthesis, a potent vasodilator with a major function in endothelial cells. ADMA is an inhibitor of the enzyme responsible for NO production; there is evidence that elevated ADMA levels are associated with endothelial dysfunction, and consequently, pre-eclampsia.[48 61] NO is an important regulator of trophoblast implantation, differentiation, motility, invasion and apoptosis.[42] Decreased levels of arginine and homoarginine in blood samples of pre-eclampsia patients reinforce the involvement of NO molecule in the aetiology of the disorder.[42 50]

Carnitine is a non-essential amino acid strongly associated with lipid metabolism. Its function is to bind to fatty acid, form acyl-carnitine and shuttle it to mitochondria, to integrate the mitochondrial oxidative metabolism.[52] The increment of carnitine might be associated to the oxidative stress observed in pre-eclampsia.[24 25 35 46 49 52]

### Lipids

Maternal serum lipids are markedly elevated in healthy pregnancies, probably a hormone-induced increase.[62] Pre-eclamptic women may have even higher levels.[63] The lipidome—the group of lipid metabolites—contains key mediators of vascular tone (sphingosine), inflammation (prostaglandins) and insulin sensitivity (free fatty acids).[24] Lipids are involved in the aetiology of pre-eclampsia either indirectly, as a substrate, or directly, as a disease mediator.[63] Lipid peroxides may cause endothelial cell activation and reinforce the action of diabetes mellitus and essential hypertension as important risk factors to pre-eclampsia.[23]

In a healthy pregnancy, the proangiogenic biochemical scenario, sphingosine-1-phosphate and its receptors are up-regulated. Ceramide levels are also increased, reflecting the trophoblastic apoptosis process, observed because of normal syncytial fusion of villous trophoblasts.[63] On the other hand, in pre-eclampsia, low levels of sphingosine-1-phosphate (angiogenic) and high levels of ceramides (proapoptotic) were found in serum analyses of third trimester pre-eclampsia cases.[26] The increase of sphingomyelin SM:18 might serve not only as atherogenic marker of pre-eclampsia development and progression, but also the marker of cardiovascular complications developed later in women who had previous pre-eclamptic pregnancies. Recent studies have demonstrated that serum elevation of sphingomyelin SM:18 positively correlates

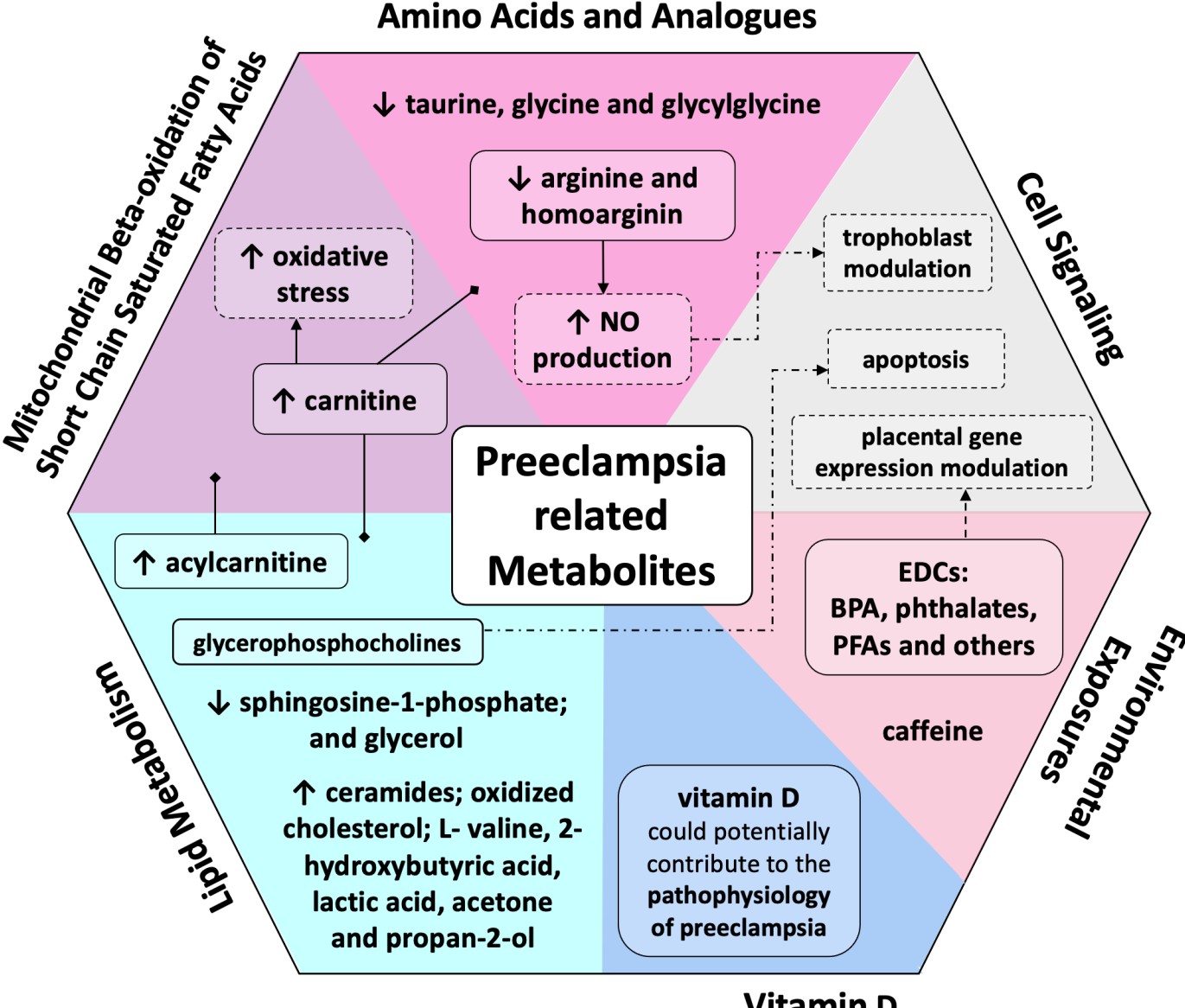

**Figure 4** Pre-eclampsia related metabolites, chemical classes and biological processes. EDC, endocrine disrupting chemical; PFAs, polyfluoroalkyl substances.

with parameters of insulin resistance and liver function in obese adults[64] and correlates with markers of NFkB activation and, thus, markers of intracellular inflammation.[64] The ceramides might be biomarkers for cardiovascular disease in later life, which points attention to its role in hypertension in pregnancy.[65] As ceramide and their synthesising enzymes (eg, ceramide synthases) have been shown to be significantly upregulated in the mouse uterus during early gestation,[44 66] the role of ceramides in the normal human pregnancy and its relationship with pre-eclampsia etiopathogenesis needs to be further explored.

Pre-eclampsia has also been correlated with high amounts of circulating reactive oxygen species. Among the consequences of this, high levels of oxidised cholesterol and acylcarnitine can be attributable to oxidation impairment of phosphocoline and fatty acid, respectively.[66]

Glycerophosphocholines resulted from cell lysis, possibly correlated with apoptosis described in pre-eclampsia. For the propanoate pathway, late-onset pre-eclampsia showed higher concentrations of L- valine, 2-hydroxybutyric acid, lactic acid, acetone and propan-2-ol.[52] Decreased glycerol level has been regarded as an important biomarker of late-onset pre-eclampsia and this may be due to its conversion to triglycerides.[57 63]

**Vitamin D**

The included studies on vitamin D showed contradictory results.[29 34 36 41 44 51] The heterogeneity of results may be attributable to the measurement of only serum 25(OH)D3 vitamin D, which is considered simplistic.[29] Interestingly, all included studies have considered low-risk women and have presented data irrespective of gestational age of pre-eclampsia diagnosis. The active

metabolite—1,25-dihydroxyvitamin D—is thought to increase vascular endothelial growth factor transcription,[67] which is correlated to placenta functioning and adaptation. Besides, vitamin D deficiency may be associated with an increment of inflammatory cytokine levels, such as TNF-alpha, and to a tendency towards T-cell activity.[68] Thus, its deficiency could potentially contribute to the pathophysiology of pre-eclampsia: abnormal placental implantation, angiogenesis, excessive inflammation, immune dysfunction.[69] Accuracy performance of vitamin D was increased when combined with angiogenic biomarkers (VEGF +sFLT-1/PlGF).[41] Additionally, all targeted studies showed lower levels of vitamin D metabolites, both in serum and urine, while untargeted LC-MS evaluation observed upregulated levels. Then, based on our results, we cannot rule out vitamin D role in predicting pre-eclampsia in high-risk population or early onset or late onset of disease. It seems there is a place for investigating its role in composing a multifactorial model of prediction.

### Environmental exposures

Knowledge of the actual contribution of endocrine disrupting chemicals, such as BPA, phthalates and others to pre-eclampsia is relevant, given that behavioural attitude can change maternal exposure to these risk factors. It is hypothesised that BPA can cross the placenta and induce degeneration and necrosis of placental cells, disturbing angiogenesis,[47] and possibly acting through estrogenic-related receptor *gama*. Phthalates can affect placental gene expression determining decreased placental growth,[64] and may also cause a proinflammatory response and increase the oxidative stress.[70] Although their role in pre-eclampsia remains elusive,[27 31 47 60] we cannot rule out their role before pregnancy: BPA, phthalates and PFASs are widely present in many products of daily life, and usually have half-lives of many years.

Another environmental factor that was pointed in our review was caffeine. Caffeine metabolism occurs in the cytochrome P450 1A2 system, and the primary metabolite of dietary caffeine (80% of caffeine by-product) is the paraxanthine. The activity of this cytochrome can be demonstrated by the ratio of paraxanthine/caffeine. During pregnancy, the primary pathway for all caffeine clearance happens in maternal cytochrome P450 1A2 system, as neither placenta nor the fetus demonstrates this cytochrome activity.[71] The decreased risk of severe pre-eclampsia with increasing paraxanthine/caffeine ratio suggests that faster caffeine metabolism may be associated with a lower risk of pre-eclampsia.[45] This is not exactly in accordance with other studies, which indicated a direct influence of caffeine levels on the risk of pre-eclampsia and failed to perform analysis of caffeine metabolism.[72] On the other hand, caffeine and its metabolites are antagonists of the adenosine A1 receptors, implicated in proximal tubular sodium reabsorption.[73] Although caffeine consumption may be associated to elevated blood pressure in the short term (≤3 hours), it does not represent a cardiovascular risk factor in the long term (2 weeks) for non-pregnant adults.[74] Future studies should consider daily intake of caffeine, which was not explored in the included study, and maternal smoking as potential bias since cigarette use induces the P450 1A2 system.[75]

### Methodological quality

Although the majority of included studies were ranked having a low risk of bias and low concerns around applicability, there is still a place for improving data reporting in metabolomics.[76] Interpreting (or using) data from bench analyses in a clinical context requires the clear description of reference standards and index tests, and timing of index test analysis. As pre-eclampsia has a complex aetiology and is a multifactorial disease, improving patient selection should be a focus of attention in subsequent studies. Better accuracy could be achieved if data from low-risk and high-risk women were to be separated, or single versus multiple gestations.

### Strengths and limitations

This systematic review can be useful as a guidance of metabolites and their performance as predictors of pre-eclampsia. We have applied a systematic review strategy in eight different electronic databases. Then, we presented possible biomarkers of pre-eclampsia that could ground future research in this area.

Unfortunately, there are limitations. The inclusion of case–control studies could add risk of bias, confounding and potential measurement error. However, we understand this is an important strategy to biomarkers selection and performance evaluation steps in metabolomics[6] in perinatal research. There was great heterogeneity among included studies. Per se, metabolomics is an overly complex process, influenced by sample collection, storage and preparation, analytical platform applied, statistical tests performed, time of day collection, fasting or non-fasting state prior the samples.[6] Thus, results presented in the studies must consider all these variables. Because of this complexity and authentic heterogeneity of metabolomics, the lack of standardisation of pre-eclampsia concept (early and late-onset pre-eclampsia) and time of sample collection represented a limitation to our study. Thus, the results could not be correlated with this important variable. Additionally, since only two studies evaluated pre-eclampsia prediction regarding fetal sex, our findings should be interpreted with caution.

### Conclusion and implications for practice

We have presented a list of 122 different metabolites and their accuracy measures, when possible. They are mainly involved with ammonia recycling; amino acid and lipids metabolism; arachidonic acid metabolism; cell signalling; galactose, lactose and nucleotide sugars metabolisms. Maternal blood seems to be the best specimen to predict early pre-eclampsia if sampled in the first trimester, and to predict late-onset pre-eclampsia in the second trimester. Future metabolomics platforms should be as

comprehensive as possible to evaluate the contributions of amino acids, peptides, fatty acid esters and carbohydrates conjugates, as pre-eclampsia biomarkers.

Although individual metabolites have low predictive measures, they should be considered in multifactorial models of metabolites alone or in combination with other biomarkers.[25 33 38 41 43 46 50 52] Metabolite signature may contribute to further insights into the pathogenesis of pre-eclampsia and to design screening tests, which may contribute to early recognising high-risk women to hypertensive disorders In future, it is important to validate citrate and carnitines as common markers between early-onset and late-onset pre-eclampsia, considering the feasibility of universal screening.[25 49 50] Additionally, since pre-eclampsia and SGA are related to abnormal deep placentation,[77] it seems plausible to study both conditions simultaneously, at least considering the metabolites found in common (sphinganine 1-phosphate, sphingosine 1-phosphate, tyrosine, vitamin D, N1, N12-diacetylspermine).[56]

**Acknowledgements** We are grateful to all members of the PRETERM-SAMBA study research group.

**Contributors** JM is the guarantor of this review. JM and DFL elaborated the protocol, searched the literature, selected studies, extracted data, assessed risk of bias, and drafted the initial manuscript. MLC has helped in study inclusion, interpreting data, examining clinical applications of findings, and revising the manuscript. GMN has helped in figures and tables elaboration and with the final version of the manuscript. JGC have supervised and approved all steps. All authors have read and agree with this submission.

**Funding** DFL (process number 88881.134512/2016-01) has awarded scholarship from Brazilian Federal Agency for Support and Evaluation of Graduate Education (CAPES). This research is part of the PRETERM-SAMBA study, which has granted sponsor from Brazilian National Research Council (CNPq) (Award 401636/2013-5) and from the Bill and Melinda Gates Foundation (grant OPP1107597).

**Disclaimer** This specific analysis received no specific grant from any sectors. Our sponsors have not intervened in any step of this study design, development, or submission.

**Competing interests** None declared.

**Patient and public involvement** Patients and/or the public were not involved in the design, or conduct, or reporting, or dissemination plans of this research.

**Patient consent for publication** Not applicable.

**Ethics approval** This is a systematic review dealing with data from articles already published and, therefore, not directly involving human subjects. It has then no need for ethical assessment and approval according to the national rules.

**Provenance and peer review** Not commissioned; externally peer reviewed.

**Data availability statement** Data sharing not applicable as no datasets generated and/or analysed for this study. All data relevant to the study are included in the article or uploaded as online supplemental information.

**ORCID iDs**
Guilherme M Nobrega http://orcid.org/0000-0001-9406-4256
Jose Guilherme Cecatti http://orcid.org/0000-0003-1285-8445

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
