## [Reviewer comments · BMJ Open]

ARTICLE DETAILS

TITLE (PROVISIONAL)	Prediction of pregnancy-related hypertensive disorders using metabolomics: a systematic review
AUTHORS	Mayrink, Jussara; Leite, Debora; Nobrega, Guilherme; Costa, Maria Laura; Cecatti, Jose

VERSION 1 – REVIEW

REVIEWER	Bernardi, J Universidade Federal do Rio Grande do Sul, Nutrition
REVIEW RETURNED	19-Sep-2021

GENERAL COMMENTS	Prediction of pregnancy-related hypertensive disorders using metabolomics: a systematic review The systematic review aimed to determine the accuracy of metabolomics in predicting pregnancy- induced hypertensive diseases. The manuscript has a careful methodology, with the previously published protocol, following the PRIMA statement and QUADAS-2 tool; however, it requires minor considerations to improve the quality of the text. Title I suggest include the word “accuracy”. Abstract I suggest include more details about result section, for example, type statistical parameters, direction of results. Strengths and limitations I suggest replace “comprehensive literature” for systematic review”. Introduction I suggest include more current references at the end of each sentence, for example, first, second and forth paragraphs. Methods I suggest explain the reason for including only cohorts and control case control studies. Results I suggest review what the authors did in the case of duplicate data. I suggest review the sentence “There were no studies with maternal hair or amniotic fluid”. I suggest throughout the text to include information in the studies cited about gestational weeks or trimesters and direction of the main findings.
---

	In the item about “Quality assessment and risk of bias” there are two final points. In table 2, I suggest standardizing “weeks” or “w” and included in legend. In figure 1, I suggest explain the manuscript removed in “identification” for “screening” phases. In figure 2, I suggest include the legend. Discussion At the beginning of the discussion, I suggest include the direction of the main findings founded. I suggest include the amount of caffeine consumption described in the study. In the item “Strengths and limitations”, I suggest replace “comprehensive search” for systematic review”. I suggest that the authors reflect on the limitations on potential sexdependent outcomes. Conclusion I suggest include implications about interventions in pregnant women. References Update access date (reference number 17).
--	--

REVIEWER	Perkins, Tony Griffith University, School of Medical Science
REVIEW RETURNED	10-Jan-2022

GENERAL COMMENTS	This is a quality submission that brings together the data from 31 studies on metabolomics and pregnancy associated hypertension. This submission makes a meaningful contribution to this field of knowledge and deserves publication. Some points of clarification require addressing and there are many typographical and grammatical errors that require fixing. 1) Some consideration should be given, perhaps in the limitations section, on sample collection. Collection of human samples is often variable in terms of time of day, season, fasting v non-fasting and these can affect the metabolome. A good example is when samples are collected for lipid analysis, usually early am with overnight fasting. This same approach needs to be applied to metabolomic studies but often is not. 2) The inclusion and exclusion criteria are confusing. For example on Page 7 it states ” We excluded studies in which blood samples were collected after diagnosis of hypertension was established, or when preeclampsia and gestational hypertension were analysed together”. Then on Page 8 “We included studies in which women with any form of pregnancy-induced hypertension were compared to pregnant women without hypertension”. This needs to be addressed. 3) Page 22 Line 16 – I would argue that preeclampsia and SGA are associated with shallow trophoblast invasion and poor placentation NOT “deep” placentation as stated. Minor Corrections P6 L14 relies on P6 L18 to identify a P10 L20 dysfunction P11 L8 NMR
--

	P15 L21 4.4 ug/L P16 L20 remove "of" P18 L13 preeclamptic women P18 L28 previous preeclamptic pregnancy P19L4 mouse not mice P19 L9 Phosphocholine
--	---

VERSION 1 – AUTHOR RESPONSE

Reviewer: 1

Dr. J Bernardi, Universidade Federal do Rio Grande do Sul

Comments to the Author:

1. The systematic review aimed to determine the accuracy of metabolomics in predicting pregnancy- induced hypertensive diseases. The manuscript has a careful methodology, with the previously published protocol, following the PRIMA statement and QUADAS-2 tool; however, it requires minor considerations to improve the quality of the text.

We kindly appreciate your time in reviewing our paper.

2. Title: I suggest include the word "accuracy".

We appreciate your comment. However, we would prefer not to include "accuracy" in the Title, since it was not possible to perform a metanalysis.

3. Abstract: I suggest include more details about result section, for example, type statistical parameters, direction of results.

We understand your suggestion and apologize for not being able to provide data on statistical procedures. We could not perform a metanalysis. However, we added the direction of results in the Abstract:

"Among the 121 different metabolites found, there were 23 amino acids and 21 fatty acids." (Page 2).

4. Introduction: I suggest include more current references at the end of each sentence, for example, first, second and forth paragraphs.

Thank you for the suggestion. We have updated some references.

5. Methods: I suggest explain the reason for including only cohorts and control case control studies.

Thank you for this suggestion. It was addressed in Methods section, by amending this sentence (Page 7, item Search Strategy).

"We intend to show the capacity of metabolomics to predict hypertensive disorders without interference of any intervention. Thus, it is absolutely necessary to guarantee the temporal relation between timing of sample (before) and outcome (after). This is the reason for including just cohort and case control studies."

Results:

6. I suggest review what the authors did in the case of duplicate data.

We apologize for being unclear. We added this sentence in the first paragraph of Results: "Regarding duplicated data, only the most recent and complete study was included."

7. I suggest review the sentence "There were no studies with maternal hair or amniotic fluid".

We apologize for being unclear. We replaced the word "with" by the word "analyzing" (Results section, 4th paragraph):

"There were no studies analyzing maternal hair or amniotic fluid."

8. I suggest throughout the text to include information in the studies cited about gestational weeks or trimesters and direction of the main findings.

9. In the item about "Quality assessment and risk of bias" there are two final points.

Thank you for this observation. We have amended this issue.

10. In table 2, I suggest standardizing "weeks" or "w" and included in legend.

We appreciate your comment. We have standardized "w" and explained in the legends.

11. In figure 1, I suggest explain the manuscript removed in "identification" for "screening" phases.

We appreciate your comment. We included a box with the number of duplicate papers excluded in this first step.

12. In figure 2, I suggest include the legend.

Thank you for your comment. All the legends are included at the end of the paper, after the References.

The legend for Figure 2 is:

"Figure 2. Characterization of study selection regarding biological samples, sampling moment and metabolomic analysis techniques applied."

Discussion

13. At the beginning of the discussion, I suggest include the direction of the main findings founded.

We apologize for being unclear. We tried to improve understanding by amending this second sentence of the first paragraph:

"In this systematic review, we have found that amino acids and fatty acids were the most common chemical subclasses associated to preeclampsia."

14. I suggest include the amount of caffeine consumption described in the study.

We have included one study on caffeine and preeclampsia (*Eichelberger et al, 10.1097/AOG.0000000000001041*). Unfortunately, the amount of caffeine consumption was not

provided in the article. This was a study with the purpose of investigating the relationship between metabolites of caffeine and preeclampsia; they did not measure the daily intake of caffeine, but only the metabolites of it. We added a sentence in the text:

“Future studies should consider daily intake of caffeine, which was not explored in the included study, and maternal smoking as potential bias since cigarette use induces the P450 1A2 system.”

15. In the item “Strengths and limitations”, I suggest replace “comprehensive search” for systematic review”.

Thank you for your comment. We have amended this section according to the Editor’s recommendations.

16. I suggest that the authors reflect on the limitations on potential sex dependent outcomes.

Thank you for this observation. We have found only two studies regarding preeclampsia prediction and fetal sex. As they evaluated different metabolites, we could not synthesize the literature properly. We have added this observations in the Results section (Summary of included studies):

“Gong et al. [28] analysed serum samples in different moments of pregnancy, but it was at 36w when N1, N12 diacetylspermine were higher in women who later present preeclampsia, especially in the case of a female fetus.”

“When the findings were analysed according to fetal sex, exposure to bisphenol and di(2-ethylhexyl) phthalate was only associated to preeclampsia in pregnancies of female fetuses.”

In the Strengths and limitations section, we added:

“Additionally, since only two studies evaluated preeclampsia prediction regarding fetal sex, our findings should be interpreted with caution.”

17. Conclusion: I suggest include implications about interventions in pregnant women.

Thank you for this suggestion. We added a sentence with this content in the Conclusion section:

“Metabolite signature may contribute to further insights into the pathogenesis of preeclampsia and to design screening tests, which may contribute to early recognizing high risk women to hypertensive disorders.”

18. References: Update access date (reference number 17)

Thank you. The update was done!

Reviewer: 2

Prof. Tony Perkins, Griffith University

Comments to the Author:

This is a quality submission that brings together the data from 31 studies on metabolomics and pregnancy associated hypertension. This submission makes a meaningful contribution to this field of knowledge and deserves publication. Some points of clarification require addressing and there are many typographical and grammatical errors that require fixing.

We kindly appreciate your time in reviewing our paper and apologize for our English errors. We have amended the text according to your suggestions.

1) Some consideration should be given, perhaps in the limitations section, on sample collection. Collection of human samples is often variable in terms of time of day, season, fasting v non-fasting and these can affect the metabolome. A good example is when samples are collected for lipid analysis, usually early am with overnight fasting. This same approach needs to be applied to metabolomic studies but often is not.

Thank you for this observation. We added a sentence with this content in the Strengths and Limitations section:

“Per se, metabolomics is an overly complex process, influenced by sample collection, storage and preparation, analytical platform applied, statistical tests performed, time of day collection, fasting or non-fasting state prior the samples.”

2) The inclusion and exclusion criteria are confusing. For example on Page 7 it states “ We excluded studies in which blood samples were collected after diagnosis of hypertension was established, or when preeclampsia and gestational hypertension were analysed together”. Then on Page 8 “We included studies in which women with any form of pregnancy-induced hypertension were compared to pregnant women without hypertension”. This needs to be addressed.

Thank you for your comment. Our main interest is to identify specific metabolites relating to each form of pregnancy induced hypertension. This is the reason why we excluded manuscripts in which more than one clinical form of hypertension (e.g., preeclampsia and gestational hypertension) were analyzed together, as the same outcome. We have amended the sentence as follows:

“We excluded studies in which blood samples were collected after the diagnosis of hypertension was established, or when preeclampsia and gestational hypertension were analysed as the same outcome.”
(Page 5-6).

3) Page 22 Line 16 – I would argue that preeclampsia and SGA are associated with shallow trophoblast invasion and poor placentation NOT “deep” placentation as stated.

We understand this term might be confusing. We have opted to maintain the terminology adopted by Brosens et al (“The ‘great obstetrical syndromes’ are associated with disorders of deep placentation. DOI: 10.1016/j.ajog.2010.08.009).

4) Minor Corrections

P6 L14 relies on

P6 L18 to identify a

P10 L20 dysfunction
P11 L8 NMR
P15 L21 4.4 ug/L
P16 L20 remove "of"
P18 L13 preeclamptic women
P18 L28 previous preeclamptic pregnancy
P19L4 mouse not mice
P19 L9 Phosphocholine

Thank you for helping us with this issue. All suggestions were amended in the main text.